# Cooperation in the face of disaster

**Marijane Luistro Jonsson** [1,2☯*], **Markus Jonsson**[3,4☯]

**1** Center for Sustainability Research, Stockholm School of Economics, Stockholm, Sweden, **2** Department of Neurobiology, Care Sciences and Society, Karolinska Institute, Stockholm, Sweden, **3** Center for Cultural Evolution, Stockholm University, Stockholm, Sweden, **4** Department of Oncology-Pathology, Karolinska Institute, Stockholm, Sweden

☯ These authors contributed equally to this work.
* marijane.luistro.jonsson@ki.se

**Data availability statement:** All experiment data and the simulation source code are available at github.com/markusrobertjonsson/condcoop. Below is our Data Sharing Plan: 1. Data Description This study uses empirical data collected during the experiments described in the paper, as well as simulation data generated through custom code. The shared dataset includes: • Raw experimental data in XLSX format. • Source code for simulations and data

## Abstract

As calamities and health crises are expected to recur and become more frequent, we rely more on cooperation to prevent similar situations and to cope with their aftermaths. However, it is not clear if, how and why people cooperate in uncertain situations where losses can result from inadequate cooperation. Through theoretical modelling, experiments and simulations, we show the behavioural patterns driving cooperation in a stochastic environment. Specifically, by introducing stochastic shocks to a threshold public goods game where one can randomly incur losses when group contributions are below a specific level, we investigate what happens to cooperation when disasters strike repeatedly. The findings show that compared to a control setting, cooperation is higher and persists when there is a risk for disasters to strike, and that this is sustained by unconditional cooperation. People give more and do not match the contributions of others, contrasting the conditionality observed in deterministic environments. In other words, we observe a contribution divergence in uncertain environments wherein some give unconditionally while others free-ride. We study three different types of uncertainty about the disaster: the probability of a disaster, additionally if it is uncertain how much cooperation is required to avoid them (threshold level), and how much losses will be incurred (impact). The results are similar in countries having different natural disaster risks, the Philippines and Sweden. Simulating for a longer time period suggests the importance of promoting unconditionality to foster sustained cooperation in facing an uncertain world.

## Introduction

Cooperation is a fundamental mechanism in the biological and cultural evolution of humanity, and people often conditionally cooperate with others [1,2]. A common definition of cooperation is when one individual pays a cost for another to receive a benefit [2], and this has been studied game-theoretically and experimentally through the standard public goods game (e.g. [3,4]). In this deterministic environment, participants face a prisoner's dilemma in deciding how much of their given endowments will be contributed to the common good [5]. This situation is often connected to but not limited to impacts of human activities to the natural environment, such as overgrazing [6,7]. Forest lands, for example, are often privately cleared for logging and farming, leading to a tragedy of the commons marked by erosion, flooding

processing scripts. • Accompanying README files describing the structure and contents of the data. 2. Data Availability The full dataset and code are publicly available in the following GitHub repository: https://github.com/\protect\penalty-\@M{}markusrobertjonsson/condcoop/ 3. Licensing and Reuse Conditions • Source Code: Licensed under the MIT License. Users are free to use, modify, and distribute the code with attribution. • Data: Licensed under Creative Commons Attribution 4.0 International (CC BY 4.0). The data may be freely shared and adapted with proper credit to the authors.

**Funding:** M.L.J.'s PhD studies were funded by the Jan Wallanders och Tom Hedelius Stiftelse samt Tore Browaldhs Stiftelse foundation, including the experiments in this study. The foundation did not play any role in the study design, data collection and analysis, decision to publish, or preparation of the manuscript.

**Competing interests:** The authors have declared that no competing interests exist.

and desertification. Theoretically, the dominant individual optimal strategy is to not contribute, because regardless of the actions of others, one will always be better off keeping one's endowments. However, if everyone resorts to this free-riding, the group suffers since everyone would have been better-off contributing, displaying that the favorable behavior for the individual is not best or efficient for the group. In time, with repeated interactions, contributions are expected to converge to non-cooperation [8]. This is supported by experimental studies, empirically revealing that a majority of the participants cooperate conditionally, as they adjust or match their low and decreasing contributions to those of others (e.g. [9,10]). Conditional cooperation is prevalent in different societies, both in stable, affluent countries [10–12] as well as in the developing world [13,14].

Cooperative behavior is however not as straightforward to predict in a stochastic environment, where disasters that can lead to losses may occur repeatedly. Human judgements are not always predictable in times of uncertainty [15], but it is nonetheless crucial to understand human cooperation in this context. As we live in the Anthropocene age, the natural environment becomes more vulnerable and uncertain, depicted by human activities crossing over planetary boundaries [16]. The environmental alterations, such as climate change and biodiversity loss, result to erratic changes in the natural systems, and eventually lead to catastrophic consequences to human systems. We already witness the direct and indirect effects of recurring hazards and disasters with the increasing cases of flash floods, forest fires, catastrophic storms, disease outbreaks, and extreme weathers, causing not only economic losses but loss of lives as well. In making these social ecological systems more robust, it has been argued that the link and cooperation between resource users and public infrastructure providers should not be ignored [17]. There are complex and nonlinear linkages between the environmental and human systems [18], but oftentimes policies addressing these systems assume deterministic settings.

This study investigates cooperation in a stochastic environment through a multi-modal approach consisting of theoretical modelling, empirical experiments and simulations that build on each other. We initially identified the equilibria predictions, and experimentally tested the models to find out what the contribution levels converge to. The empirical findings were subsequently used as a basis for a simulation of interactions for a longer period of time to analyze identified mechanisms driving the results. The interplay between theoretical modelling, experimental work, and simulations allows this study to address the limitations of each of the methods involved. For instance, the empirical data can shed evidence on equilibrium selection to theoretical analyses with multiple equilibria, while the simulations replicated empirical conditions to study the effects of specific combinations of agent strategies for a longer time period, which is not possible to manipulate in experimental settings. The next section provides a review of the previous studies that have provided a foundation for the theoretical modelling.

## Cooperation in a stochastic environment

One way in which cooperation has been studied in a stochastic environment is through threshold public goods games (TPGG), also called *provisional point mechanisms* [19,20], *step level public goods* [21,22], *discrete public goods game* [23], *collective risk social dilemma* [24,25], or *climate public goods games* [26]. In its standard form, there is a collective target or threshold that must be reached for the provision of the public good or the avoidance of a public bad (i.e. collective loss). Here, there is an incentive for everyone in the group to contribute enough to reach the threshold to avoid the risk of a collective loss, but at the same time, there is temptation to defect or free-ride on the contributions of others, as in the standard public goods

game. The best outcome for everyone is any contribution combination that exactly meets the threshold (even if contributions are not fairly distributed among the participants). The presence of multiple equilibria in a single interaction, including zero contributions, makes it unclear if cooperation, with repeated interactions, will arise and persist or converge to the non-cooperating behavior.

Theoretical studies using the TPGG framework, both taking game theoretic and evolutionary dynamics perspectives, have attempted to model under which conditions the cooperation equilibrium emerges when there is a risk for loss in repeated interactions. Some of the identified factors leading to successful group cooperation are reciprocity [27], high value of the public good [23], high threshold uncertainty [28], high risk for a catastrophic event [29], small group size under high risk [30], intermediate feedback of performance and risk for loss [31], big initial endowments [32], and when participants care for the future [33]. Although these theoretical studies reveal valuable insights, they are based on predictions on how hypothetical agents will behave to meet the cooperative equilibrium, prompting the need for complementary empirical studies to provide evidence (e.g. [33]).

Empirical TPGG studies provide rich empirical evidence but are quite fragmented, having multiple experimental design variants. They have identified different conditions and interventions making cooperation successful, such as lower threshold [21], homogeneous groups [22], and the presence of refund guarantees [20,34] and communication [35]. More recent studies, framed in the context of global cooperation, climate change or environmental conservation, similarly use the collective risk dilemma setting to investigate the avoidance of environmental disasters under different uncertainties. In general, cooperation is more successful with higher risk on the occurrence of a loss [24], lower uncertainty on the threshold level [36] and remains the same when the consequences or impact are known [37]. Conditions that enhance cooperation include the provision of intermediate targets [25], expert information and non-anonymity of contributions [26], as well as communication of commitments [37]. Conversely, cooperation decreases with inequality [22,38], when the benefits of avoiding the catastrophe are low [37], and if the rewards of cooperation is delayed into future [39]. In the same manner, variants of common pool resource and appropriation game settings similarly study uncertainties surrounding cooperation by refraining from overharvesting resource or moving funds to prevent drastic collapse. The studies likewise find that factors such as communication [40], voting [41], slow thought processes [42], low opportunity costs [43], as well as individual optimism [43] enhance cooperation when there is a chance for resource collapse or losses. Although the extant experimental studies can shed some light on factors affecting cooperation, the studies often entail a single decision interaction (e.g. [35–37]), or if there are repeated interactions, there is only a single risk of disaster at the end of the game (e.g. [26,38]) or only one of the rounds is randomly chosen to be compensated (e.g. [44]). The variation in the experimental designs makes it difficult to compare results of various types of uncertainties and complement existing theoretical studies.

To provide more suitable empirical evidence that can inform related theoretical models, and relevantly address the increased frequency of a wide range of disaster events, this study makes a series of theoretical analyses, experiments, and simulations to study the effects of uncertainties to cooperation. It seeks to investigate if cooperation persists under different types of uncertainties, and identify possible underlying mechanisms driving it. In contrast to other studies, this study does not intend to model the dynamics of a specific environmental resource system with risk for a total system collapse (i.e. the game ends) but presents a general stochastic environment to study how human cooperate with a constant risk for repeated losses, and the interaction remains. We focus on the distinct effects of three common types of uncertainties surrounding a possible disaster: when we do not know when it will happen

(timing), how much loss will be incurred (timing+impact) and which cooperation level will avoid it (timing+threshold).

## Experiment design and theoretical predictions

To investigate how people cooperate in a stochastic environment, we compare a control group (*Control*) in a deterministic setting (standard public goods game), and different treatments depicting cooperation in stochastic settings (threshold public goods games). We initially assign participants randomly into groups of four that interact with each other throughout the experiment that lasts for 20 rounds, but the number of rounds is undisclosed to the participants. In each round, each participant is given 20 monetary units to allocate between the public pot and their individual account (i.e. what they keep for themselves). The allocation is in whole units $(0, 1, \ldots, 20)$. The total contribution to the public pot is multiplied by 1.6 and added to the group account.

After making the contribution decision in each round, participants get a summary information of how others contributed (presented in random order), what they earned in that round, and the current balance in their individual and group accounts. At the end of the experiment, the balance in the group account is divided evenly among the four participants, and the total earnings for each participant (the private account and their share of the group account) are converted to real money.

In *Control*, there is no risk for losses and everyone gets their full income from the individual and group accounts at the end of their interactions. A single-interaction equilibrium analysis of the experiment groups shows that contributions in *Control*, with four participants and 21 strategies, are expected to converge to the single, dominant zero-contribution equilibrium.

In the treatment groups, there is a probability of a random check in each round, and if the group contribution does not meet a certain level, losses in earnings are incurred. Different treatments focus on different parameters, such as the probability of a check, the impact of a failed check, and the threshold level. If there is a check, the participants get a red/green screen informing them if the group failed/passed before they get the updated results and summary information about the account balances. We specifically have these treatment groups:

- *10P*: There is a 10% probability of a check in each round, and if the group contribution is below a threshold level of 75% of total endowments in the group (i.e. 60 monetary units), the cumulative earnings in both the individual and the group account will be reduced to zero;
- *40P*: The same conditions as *10P* but there is a 40% probability of a check;
- *Impact*: The same conditions as *40P* but if the threshold is not met, there is equal probability (i.e. 1/3) that the individual account, the group account, or both accounts are reduced to zero;
- *Level*: The same conditions as *40P* but the threshold level in each round can be any integer value in a certain range (i.e. 50 to 70 units), randomly chosen with equal probability. In each round, the threshold level for that round is revealed after contributing.

Fig 1 shows an overview of the different experiment groups, and details of the experiment design and implementation can be found in S1 Appendix in the supporting information. Additional information regarding the ethical, cultural, and scientific considerations specific to inclusivity in global research is included in S4 Checklist.

In these treatments, the equilibria consist of the zero-contribution profile $(0, 0, 0, 0)$ and the group contribution combinations that add up to 60 units (70 units for *Level*). Details of

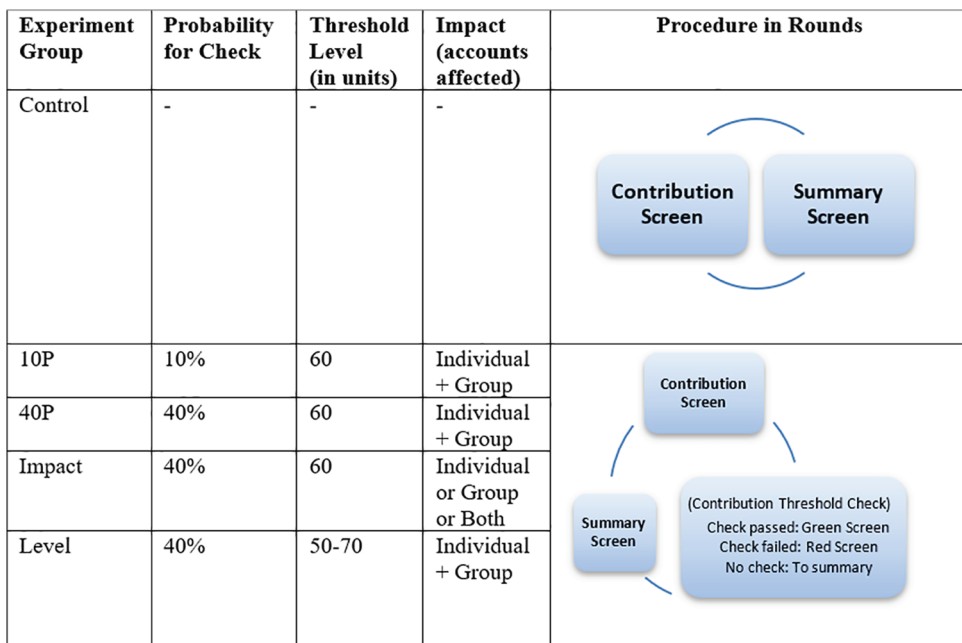

**Fig 1. Overview of the experiments.**

the equilibrium analyses for these different treatment groups are given in S2 Appendix in the supporting information. This means that we cannot a priori predict what contributions will converge to – either to a cooperative or to a free-riding regime. Various theoretical models have made predictions but they build on assumptions such as how people may discount the future (e.g. [32]), which might change alongside how people experience the disasters. Therefore, we look into and rely on empirical evidence to study which equilibrium will be most likely taken in the various experimental groups. The experiments were conducted to a total of 884 participants in Sweden and the Philippines, countries having diverging disaster risk exposures [45], to add to the external validity of the results.

## Experimental results

In statistically analyzing if there are differences among the treatments, we used OLS regressions, with standard errors adjusted for clustering on groups and participants.

### Contributions

**Effect of uncertainty.** Our results show that contributions are 27% higher in the face of disasters than in the absence of it (Fig 2A). In 58% of the threshold checks, the groups succeeded in having adequate cooperation and avoided the disaster. The contributions in the treatment groups also increased throughout the rounds, while it decreased in *Control* (Fig 2B).

This pattern of not only higher but also increasing contributions is significant and consistent for both countries and for the different types of uncertainty treatments (Figs 3A and 3B), giving evidence that the interactions end up in a cooperative regime in the face of disaster.

That contributions are higher when there is a risk of losing earnings (unless a contribution threshold is met) is both intuitive and consistent with the Nash equilibrium analysis in

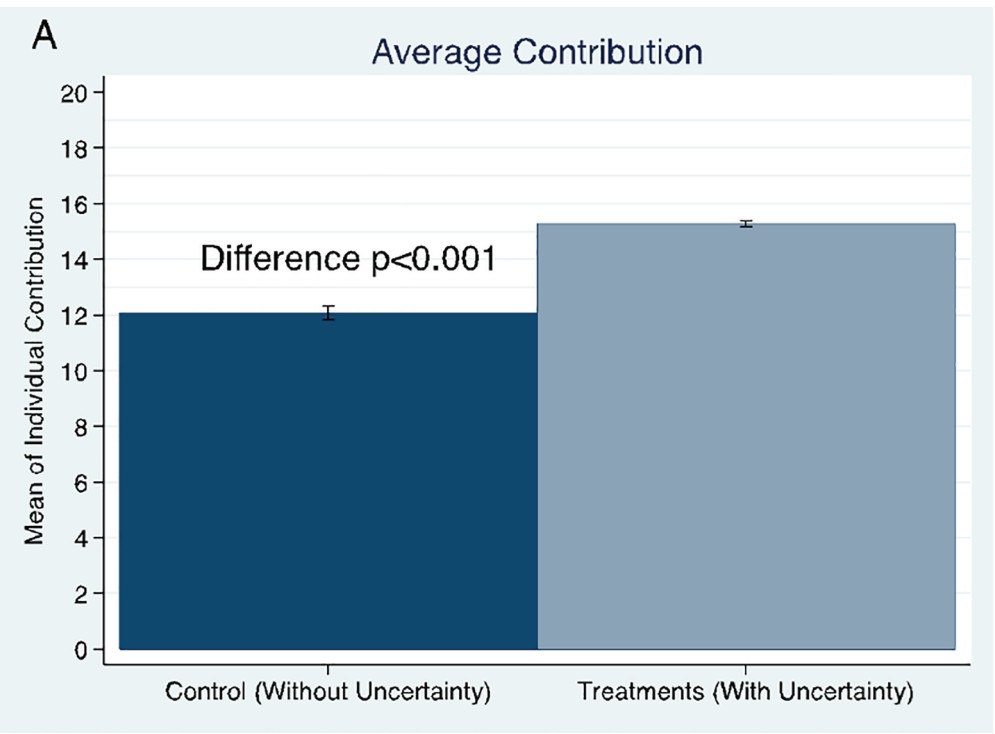

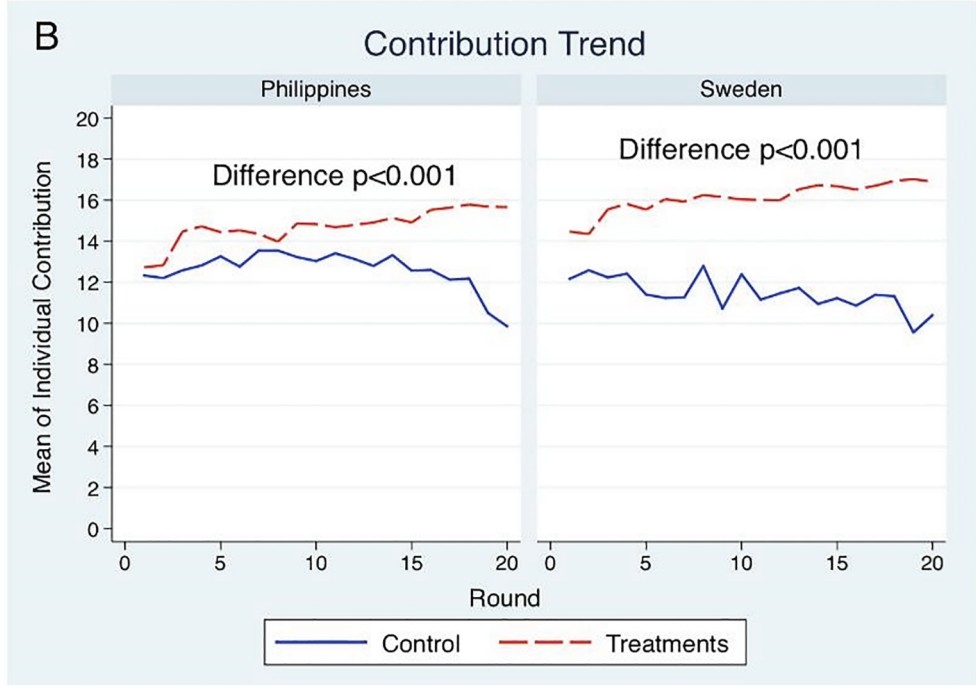

**Fig 2. Contributions: (A) Mean of individual contributions to the public good, with 95% confidence interval and (B) mean contribution over time for each country.**

S2 Appendix. However, the extent to which contributions increase in the presence of a disaster risk naturally depends on the participant's risk-willingness. We see evidence of this in the Contextual differences section below.

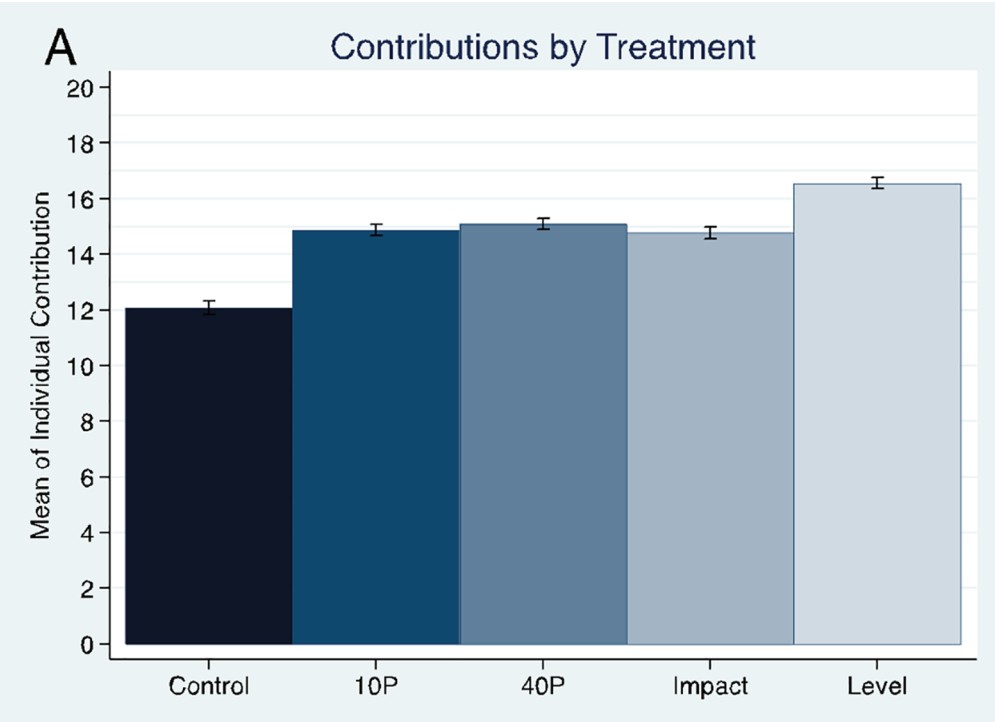

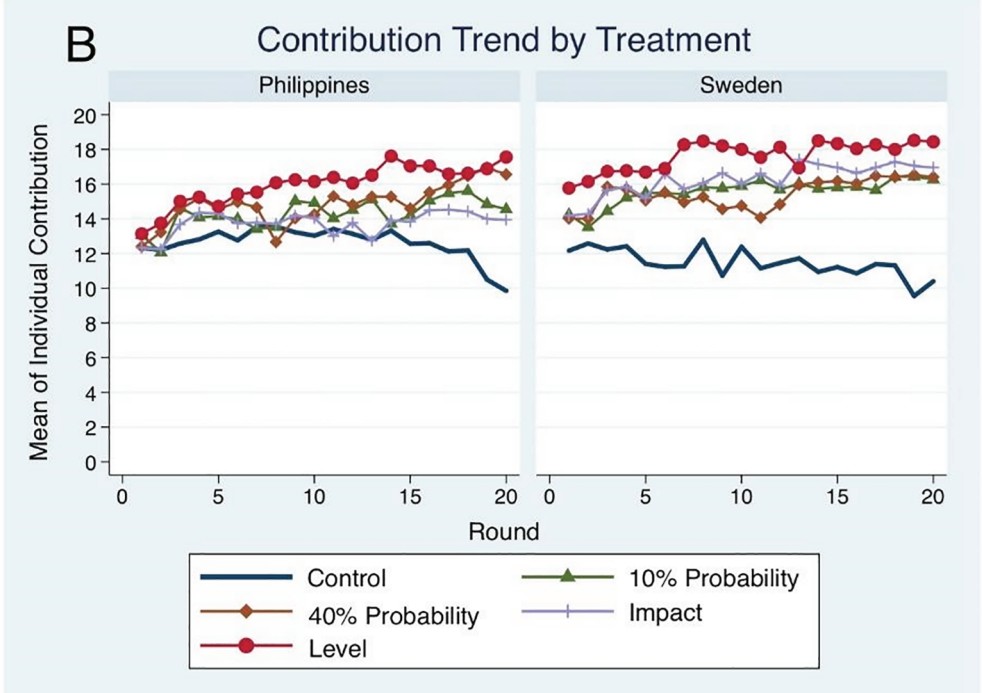

**Fig 3. Contributions of different experiment groups: (A) Mean of individual contribution to the public good, with 95% confidence interval, and (B) mean contribution over time.**

**Effect of different uncertainty types.** Focusing on the specific results for the different types of uncertainties, the ensuing patterns were exposed. There was no significant difference between contributions in *10P* and *40P* in either of the countries, showing that the existence of a possible loss, regardless of the probability level, may induce cooperation. To fur-

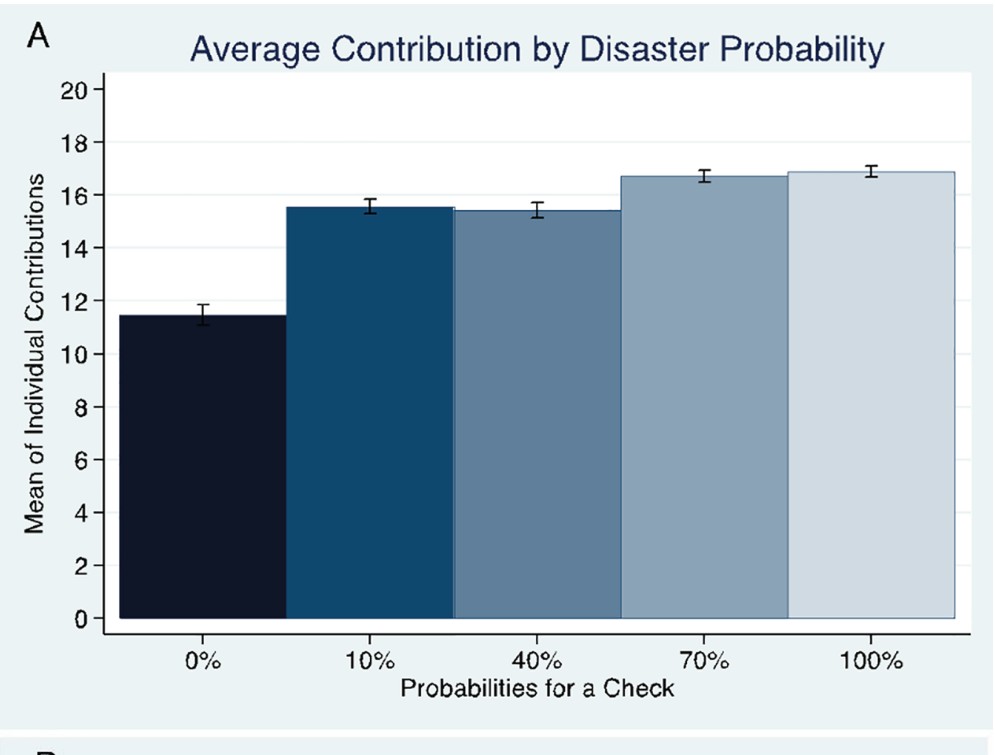

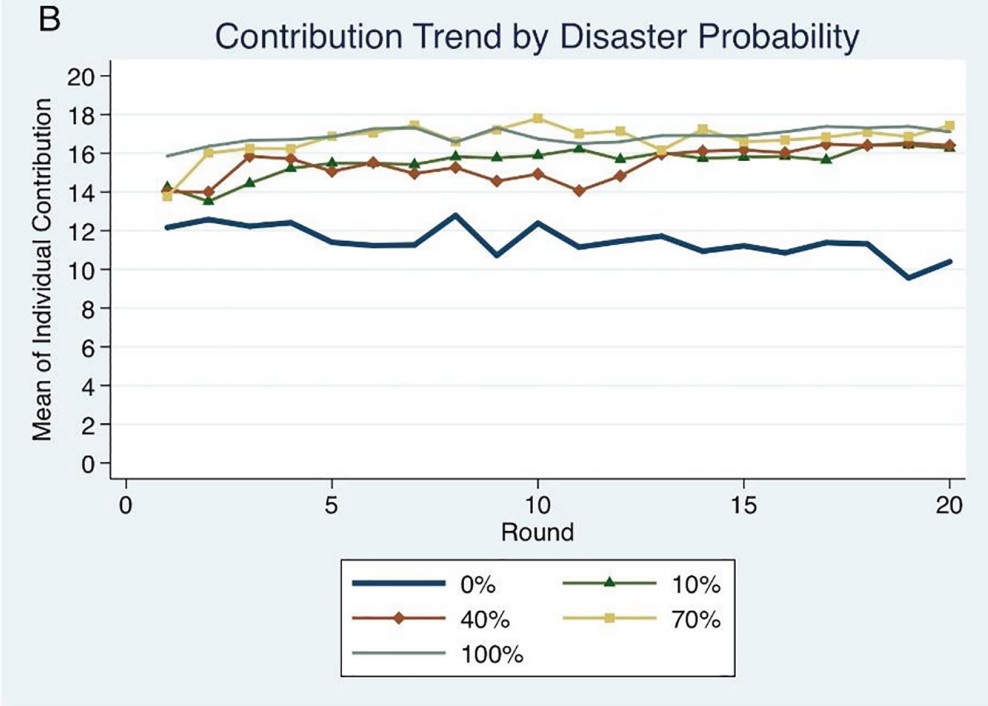

**Fig 4. Contribution given different disaster probabilities: (A) Mean of individual contribution to the public good at different probability levels for a check to happen in each round, with 95% confidence interval and (B) mean contribution over time.**

ther explore the effects of this probability, additional experiments for 70% and 100% probability were conducted in Sweden. The findings in Fig 4 show that contributions generally increase in a stepwise manner, with a significant increase between 40% and 70%, but not between 70% and 100%. The non-difference of contributions in the range of 10% to 40%, and 70% to 100%, most likely reveal the heuristics people make in their probability estimation [15]. The additional uncertainty in *Impact* did not lead to a significant difference in contribution levels compared to *40P* while the additional uncertainty in *Level* resulted in significantly higher contributions than in *40P* and *Impact*, supporting the theoretical predictions. Participants level up their contributions to the upper limit of the uncertain threshold level interval, converging to contribution combinations of 70 units.

## Contextual differences

For contextual difference, those with less exposure to real-world disasters (Swedish participants) are faster to cooperate with others in the laboratory, contributing more when the loss is not only real but also when looming. In isolating rounds wherein the threshold checks have not yet occurred (i.e. rounds before the first check), the findings show that the Swedish participants gave significantly higher contributions in all the treatment groups than *Control* ($p < 0.001$, SE = 0.964). This was not the case for the Filipino participants ($p = 0.120$, SE = 0.941). Moreover, contributions in the first round in the treatment groups were significantly higher in Sweden, see Fig 2B. These results suggest that those who are more exposed to higher disaster risks initially take more risks, procrastinating higher contributions until the checks become a reality. In the questionnaire given at the end of the experiment, the Swedes reported lower risk-willingness compared to Filipinos ($p < 0.001$).

## Responses to checks

In investigating how people behave behind the big picture of sustained cooperation, the following results prevail.

After experiencing a check where the group failed to meet the threshold, the total group contribution in the immediate round after the check does not significantly change (Sweden $p = 0.469$; Philippines $p = 0.397$). However, it then increases in the succeeding rounds, and eventually plateaus around the threshold (Fig 5A). In particular, it drastically increases in the second round after the check (Sweden $p < 0.001$; Philippines $p < 0.001$), and from the second to the third round (Sweden $p = 0.006$; Philippines $p < 0.001$). The succeeding changes between consecutive rounds tapered and were not significant any longer.

Moreover, group contributions in "almost-made-it" cases (i.e. 10% or less below the threshold level) drastically decreased in the immediate round after the check, then eventually increases, see Fig 5B. This depicts that close-call events can result in riskier decisions [46].

Both of the abovementioned results (where contributions are not increased after a failed check) is consistent with erroneous heuristics used when assessing the risk of an uncertain event, in particular the "Gambler's fallacy" – the belief that the probability of a check in the current round is lower if there was a check in the previous round [47,48].

## Conditionality

We analyzed if individuals reciprocate the contributions of others, as what previous studies have empirically found in a deterministic setting (e.g. [10,13,49]). Intuitive reciprocation, as opposed to deliberative strategies, is argued to be the driving force behind cooperation [27].

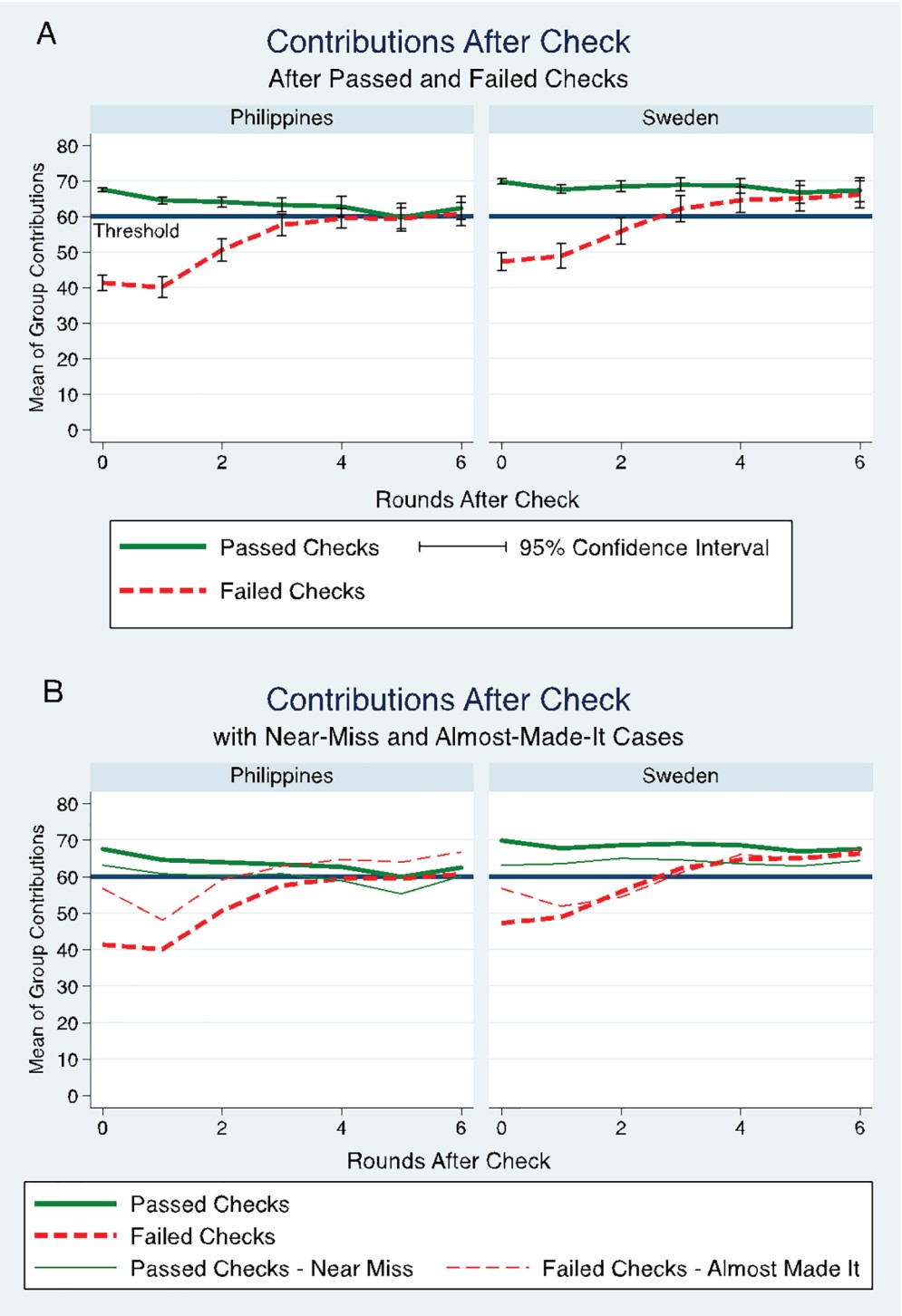

**Fig 5. Contributions after a check: (A) Mean group contribution by rounds after a check, with 95% confidence interval in groups which passed and failed the checks, and (B) for near-miss and almost made-it cases.**

We investigated if this is still the case in a repeated interaction and stochastic setting. Conditional cooperation, and its corollary unconditional cooperation, is empirically measured in this study in two ways.

Firstly, we analysed the data using a statistical-type classification algorithm that calculates each subject's linear conditional-contribution profile (LCP) as a basis of classifying the participants [50]. The LCP is the ordinary least-squares regression line of a participant's contribution on the mean contribution that he/she observed immediately before making the contribution (in our case, the average contribution of the other three participants in the previous round). The intercept of the LCP gives a measure of the subject's willingness to cooperate even if other members do not, while the slope measures the subject's responsiveness in the direction and magnitude of others' contribution. If the subject's LCP lies only in the area below 50% of the endowment level (10 units in our case), one is considered a "Free-Rider" (FR); if the LCP lies only in the area above 10 units, one is considered an "Unconditional Cooperator" (UC); and if the LCP has a positive slope and lies both above and below 10 units, one is considered a "Conditional Cooperator" (CC) (or *Reciprocator* in other studies' terminology). Subjects with LCP lines not fitting into any of these criteria are classified as "Uncategorized". Fig 6 shows a typical LCP line for each category.

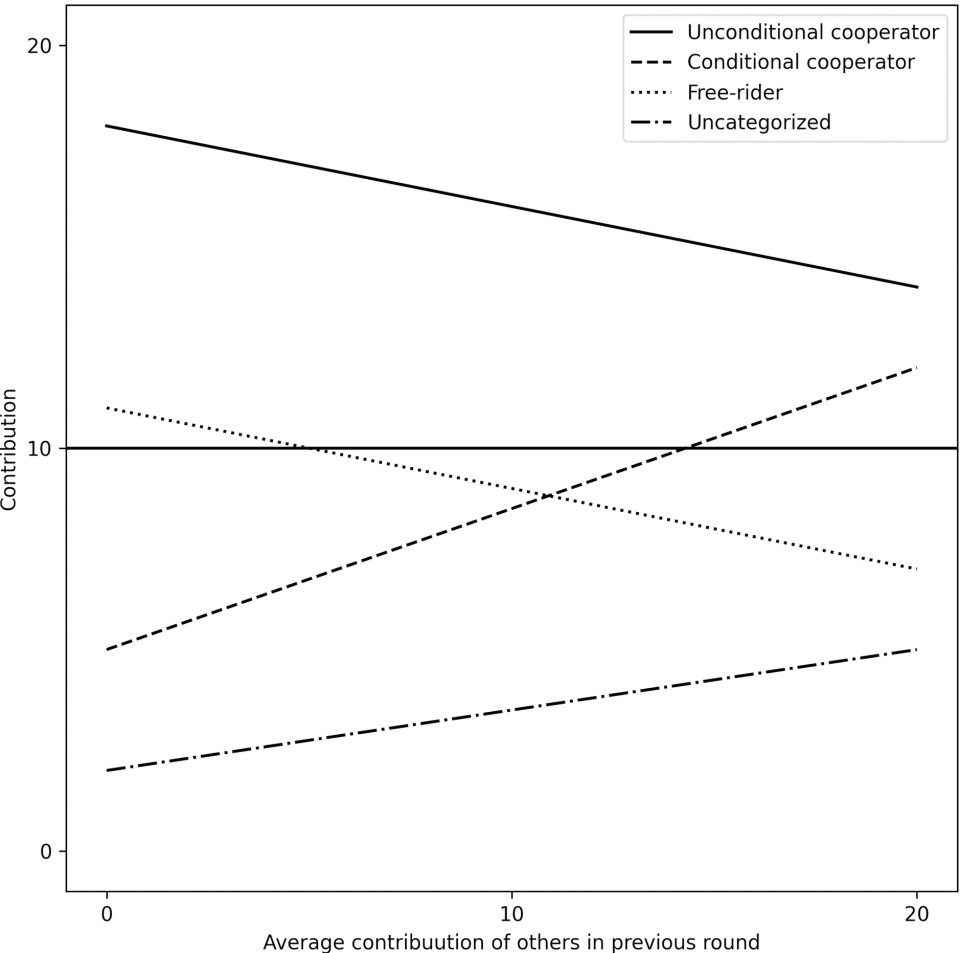

**Fig 6. Typical Linear Contribution Profiles (LCP) for the different player types: LCPs of Unconditional Cooperators are always above 10, Free riders are always below 10, Conditional Cooperators crosses 10 and has a positive slope.**

We find there were fewer CC in the treatments compared to *Control*. The distribution of player types in *Control* is similar to what earlier studies have found, where a majority behaved as CC (60%), followed by UC (24%), FR (11%), and Uncategorized (5%). On the other hand, in the treatments, the distribution shifts to more UC (56%), and fewer CC (36%), FR (4%) and Uncategorized (4%).

Secondly, we analyzed the extent of conformity, and the direction to which the participants conform. Similar to previous studies, we investigated whether people increased, decreased or did not change their contribution, depending on if they were above, below or equal to the group average [50–52]. We find that the contributions in the treatment groups did not decrease/increase as much as in *Control*, confirming that the tendency to conform is lower in the treatments.

In looking into how participants contributions ranked compared to others, we additionally find similar evidence that conditional cooperation weakens in a stochastic environment, compared to a deterministic one. In this method, people's contribution decisions are shaped based on their perception on how they deviate from the group, thus, one can conform towards the direction of the group. See S3 Appendix in the supporting information for conformity analyses.

## Equlibrium selection

The Nash equilibrium analysis in S2 Appendix in the supporting information for the treatment groups *10P*, *40P*, and *Impact* with threshold level 60 predicts convergence to a number of contribution combinations where the total group contribution is 60. Empirically, isolating these three treatments and consolidating both countries, the average contributions indeed show an approximate convergence to a level slightly above 60 units, as seen in Table 1.

To investigate which of the Nash equilibria with group contribution 60 was selected, in particular whether the symmetric Pareto-optimal equilibrium (15-15-15-15) was selected, we computed the distances between this and the group contribution combinations. This was done using a four-dimensional Manhattan metric. The results show that group contributions were far from the symmetric equilibrium. This is however expected as we have established the presence of unconditional cooperation in these treatments in the section Conditionality. In other words, there is a divergence characterized by the presence of *both* free-riders and high-contributors in the uncertainty treatments. In the presence of free-riders, the high-contributors are categorized as unconditional cooperators.

## Simulation results

Given the experimental findings, simulations were made to investigate how groups with different constellations of player types would fare in the long run, given the identified contribution patterns found in the experiments (See [53] for the source code.). For each player type, we used the average of the observed LCP lines to compute the contribution in each round based on the other participants' contributions in the previous round. We also used the average initial contributions for each player type to determine the contribution in the first round.

**Table 1. Group contributions in *10P*, *40P*, and *Impact* over round number.**

| Round | 1 | 2 | 3 | 4 | 5 | 6 | 7 | 8 | 9 | 10 |
|---|---|---|---|---|---|---|---|---|---|---|
| Contribution | 53.0 | 52.4 | 58.8 | 59.9 | 58.9 | 59.7 | 58.2 | 57.2 | 59.7 | 59.5 |
| Round | 11 | 12 | 13 | 14 | 15 | 16 | 17 | 18 | 19 | 20 |
| Contribution | 59.1 | 59.3 | 61.0 | 60.6 | 60.3 | 62.0 | 63.0 | 64.0 | 63.4 | 62.7 |

Fig 7 shows varying proportions of UC (*x*-axis), where the CC/FR ratio is fixed to the empirically found value 215/21 = 10.2. With the resulting distribution of player types, a population of 4000 individuals was divided into 1000 groups of four where each group member was assigned a player type at random with probabilities from this distribution. After 200 rounds the simulation was terminated and the converged group contribution $g$ was compared to the threshold level 60. If $g < 60$, the group is considered unsuccessful, otherwise successful. The proportion of successful groups in the population is then used as a measure of the population's success (*y*-axis). The empirical value (proportion 0.56 unconditional cooperators) is marked in Fig 7.

We find a nonlinear increase of population success with respect to proportion UC. In other words, a population gains more (in terms of number of successful groups) by increasing the proportion of UC, compared to what the population loses by decreasing the proportion of UC.

## Discussion and conclusion

The various experimental and simulation findings of this study jointly show that in the long term, cooperation can persist in the face of disasters, It also exposes the role that unconditional cooperation plays in this process. People generally chose to cooperate and avoid a disaster given a small probability of the disaster, and where the required threshold to avoid the disaster is unknown (level), as well as when the consequences of the disaster is unknown (impact). These findings give supporting evidence to earlier studies postulating how the collective dilemmas do not necessarily have to end in a tragedy [33].

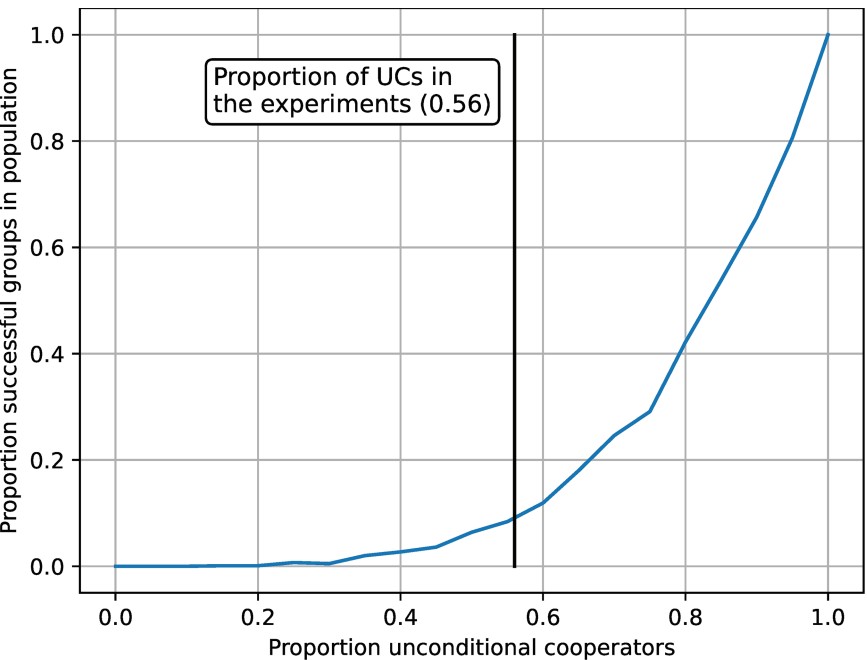

**Fig 7. Proportion of successful groups against the proportion of unconditional cooperators: Results of simulated interactions under treatments conditions for 200 rounds, based on the average empirical LCP lines per player type. The vertical black line depicts the empirical proportion of UC in the experiments.**

This study also finds that unconditional cooperation increases in these uncertain environments compared to a certain one. With the definition of unconditionality used in this paper, this increase is explained by the fact that there is both low-contributing individuals and high-contributing individuals present in these environents, giving rise to the unconditional cooperation found in this study. If all individuals matched one another's sufficiently high contributions to avoid the tragedy of the commons in the face of disasters, there would be no unconditional cooperation to measure. However, the reality of free-riding behavior gives rise to a contribution divergence and calls attention to the importance of unconditional cooperation.

Uncovering unconditional cooperation behavior is merely scratching the surface of human behavior in uncertain times, as other related mechanisms or deep-seated factors might be involved. In this study unconditional cooperation is conceptualized as the high contributions of people do not match the low contributions of others, which can stem from various reasons not necessarily limited or equated to altruism or pro-sociality. For instance, given the experimental set-up involving multiple rounds, one can interpret the behaviour as a form of learned generosity, or can alternatively result from loss avoidance. Future studies can probe more into the concept of unconditional cooperation. Nonetheless, as this study uncovers the importance of unconditional cooperation, it leaves implications not only for science to explore, but for management and policy to promote unconditional behavior. With the increasing uncertainty in the real world, it is difficult to assume that people will cooperate unconditionally, especially since humans have been used to conditionally cooperate. In the stable Holocene age, we have developed a tit-for-tat behavior to survive, and we need to be aware that others will continue to reactively cooperate conditionally or free-ride in the face of disaster.

Given that there will always be people who will free-ride, how do we then go forward promoting unconditional cooperation? The experimental settings in this study can provide some insights for conditions. In the experiments, the participants were informed and reminded of the probabilities of a disaster when they made their decisions, giving an insight how information dissemination can be important to people's capacity and psychology to cooperate. Humanity cannot rely on the looming risk for disasters alone to push cooperation, but the awareness and reminder of it can aid people in making unconditional cooperative decisions. The findings also reveal that being informed of an even low probability of a disaster can make people cooperate in the same way as a moderate one. The information nudge can help us prepare and act according to what is best for us, rather than become unnecessary victims due to ignorance, overconfidence, knowledge resistance or denial of disasters. It should be noted that people perceive information on disasters differently, and their experience of disasters and emotions play an important role. In the study, some people instantly cooperate with the presence of a threat of a loss, while others only cooperate after "getting burned". Thus, exposure to disasters can give variations to how people immediately respond to the call to cooperate, indicating the importance of spreading information effectively.

Moreover, the findings imply the importance for various institutions to structurally promote unconditional cooperation, and not just to rely on individual responsibility to encourage altruistic acts. There are social factors beyond the scope of this study, which future studies can investigate, that can have the reverse effect to cooperation in the face of disasters (e.g. inequality). Thus, considering the vulnerable state of the world with increasing polarization, there is an urgent need to establish a structural change in mindset among governing systems and institutions for bottom-up unconditional cooperation to arise in its various forms (e.g. following health restrictions, using less fossil fuels or compensating for its use). Currently, institutional structures are still designed to function in a stable environment and based on norms of reciprocity and conditional cooperation, reflected for instance by repeated failed global

cooperation agreements, which warrants transformation. Although studies on unstable environments such as this one shows the positive effects of uncertainty, the real world have other factors that can outweigh such effects. This presents an urgency for policy and management, in various levels, to create and cultivate conditions conducive for unconditional cooperation as we face more disasters.

To conclude, cooperation has been and still is a keystone of humanity's survival in times of disasters. The key lesson is that we are indeed capable of cooperating, but it is important for an unconditional cooperation mindset to dominate in order to successfully hurdle over thresholds in unstable environments. This study shows how some conditions, such as providing information about the possibilities, impact and prevention level of disasters, no matter how uncertain they are, can provide provisions for unconditional cooperation to prevail. This demonstrates the importance of transparency, effective communication, education, and transforming institutional structures that can give way to forming collective ethos encouraging unconditional cooperation. We eventually realize that we need to think of and cooperate with others to overcome disasters, regardless of the past, to save ourselves in the future. Disasters are continuously happening – remotely for some and immediate for others, and in this uncertain world where disaster can strike, there is a need for unconditionality to flourish for human cooperation to thrive.

## Supporting information

**S1 Appendix. Materials and methods.** This file contains the details of the experiment design and implementation.
(PDF)

**S2 Appendix. Nash equilibrium analyses.** This file contains the Nash equilibrium analyses of the different experiment groups.
(PDF)

**S3 Appendix. Conformity tests.** This file contains an analysis of how the participants conform to the rest of the group.
(PDF)

**S4 Checklist. Inclusivity in global research.** This file contains information regarding the ethical, cultural, and scientific considerations specific to inclusivity in global research.
(PDF)

## Acknowledgments

We acknowledge the staff of the computer laboratory at the Computational Science Research Center (CSRC), University of the Philippines Diliman, for their administrative and technical support during the experiments conducted at their facility. We also thank Alexander Funcke from the Centre for Cultural Evolution (CEK), Stockholm University, for his technical assistance in the CEK lab.

## Author contributions

**Conceptualization:** Marijane Luistro Jonsson.

**Data curation:** Marijane Luistro Jonsson.

**Formal analysis:** Marijane Luistro Jonsson, Markus Jonsson.

**Investigation:** Marijane Luistro Jonsson.

**Methodology:** Marijane Luistro Jonsson.

**Project administration:** Marijane Luistro Jonsson.

**Software:** Markus Jonsson.

**Visualization:** Marijane Luistro Jonsson, Markus Jonsson.

**Writing – original draft:** Marijane Luistro Jonsson.

**Writing – review & editing:** Marijane Luistro Jonsson, Markus Jonsson.

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
