## [Decision Letter · Decision Letter 0]

1 Dec 2024

PONE-D-24-39170Cooperation in the face of disasterPLOS ONE

Dear Dr. Luistro Jonsson,

Thank you for submitting your manuscript to PLOS ONE. After careful consideration, we feel that it has merit but does not fully meet PLOS ONE’s publication criteria as it currently stands. Therefore, we invite you to submit a revised version of the manuscript that addresses the points raised during the review process.

**ACADEMIC EDITOR: **Two reviewers have provided constructive feedback. They both agree the paper has good merit and addresses a highly relevant issue in behavioural evolution. There are some suggestions they have provided for making further improvements to the paper. Please take them carefully into account when preparing the revision version. 

We look forward to receiving your revised manuscript.

Kind regards,

The Anh Han, Ph.D.

Academic Editor

PLOS ONE

Journal Requirements: When submitting your revision, we need you to address these additional requirements. 1. Please ensure that your manuscript meets PLOS ONE's style requirements, including those for file naming. The PLOS ONE style templates can be found at https://journals.plos.org/plosone/s/file?id=wjVg/PLOSOne_formatting_sample_main_body.pdf and https://journals.plos.org/plosone/s/file?id=ba62/PLOSOne_formatting_sample_title_authors_affiliations.pdf 2. Please include a complete copy of PLOS’ questionnaire on inclusivity in global research in your revised manuscript. Our policy for research in this area aims to improve transparency in the reporting of research performed outside of researchers’ own country or community. The policy applies to researchers who have travelled to a different country to conduct research, research with Indigenous populations or their lands, and research on cultural artefacts. The questionnaire can also be requested at the journal’s discretion for any other submissions, even if these conditions are not met.  Please find more information on the policy and a link to download a blank copy of the questionnaire here: https://journals.plos.org/plosone/s/best-practices-in-research-reporting. Please upload a completed version of your questionnaire as Supporting Information when you resubmit your manuscript. 3. Please review your reference list to ensure that it is complete and correct. If you have cited papers that have been retracted, please include the rationale for doing so in the manuscript text, or remove these references and replace them with relevant current references. Any changes to the reference list should be mentioned in the rebuttal letter that accompanies your revised manuscript. If you need to cite a retracted article, indicate the article’s retracted status in the References list and also include a citation and full reference for the retraction notice.

**Additional Editor Comments:**

Two reviewers have provided constructive feedback. They both agree the paper has good merit and addresses a highly relevant issue in behavioural evolution. There are some suggestions they have provided for making further improvements to the paper. Please take them carefully into account when preparing the revision version.

Reviewers' comments:

Reviewer's Responses to Questions

**Comments to the Author**

1. Is the manuscript technically sound, and do the data support the conclusions?

Reviewer #1: Yes

Reviewer #2: Yes

2. Has the statistical analysis been performed appropriately and rigorously? 

Reviewer #1: Yes

Reviewer #2: I Don't Know

3. Have the authors made all data underlying the findings in their manuscript fully available?

Reviewer #1: Yes

Reviewer #2: Yes

4. Is the manuscript presented in an intelligible fashion and written in standard English?

Reviewer #1: Yes

Reviewer #2: Yes

5. Review Comments to the Author

Reviewer #1: This manuscript explores what happens to cooperation

when disasters strike repeatedly using a threshold public goods game with stochastic shocks. To tackle this issue, they perform subjective experiments, mathmatical analysis, and simulations.

Their methods jointly show that cooperation is higher and persists when there is a risk for disasters to strike, and that this is sustained by unconditional cooperation. This is explained by a contribution divergence in uncertain environments wherein some give unconditionally while others free-ride. The key lesson by this study is that we are indeed capable of cooperating, but it is important for an unconditional cooperation mindset to dominate in order to successfully hurdle over thresholds in unstable environments.

I agree with its publication because this manuscript has suitable research questions, methods, results, and presenations basically. I have some minor points which would be helpful for the revision.

1) I think one of key elements of this study is to test the three different types of uncertainty about the disaster (the probability of a disaster, threshold level, and impact). Therefore, I believe it is important to show the influences of them on the experimental pereformances. Although in Results, the authors mentioned contributions, contextual differences, responses to checks, and conditionarlity, the summary (figures) of results, especially the impact on the three different types of uncertainty may be helpful for readers to understand.

2) The experiments were conducted during various periods in 2012-2013. I wonder why it took for more than ten years to submit this manuscript. Would you explain the situation briefly?

3) Their simulation results support for their experiments. Although the source code is avalible, I don't understand the detailed setting of the simulation. Please add the explanation.

Reviewer #2: The authors studied the change of cooperation patterns in the presence of uncertainty, represented by various types of probabilistic risks. Overall, this is an experimental study of human behavior, and I cannot argue with empirical observations. I believe that their observations are worth reporting, so I recommend its publications with a couple of minor comments.

First, in Abstract, the authors say that ``This is explained by a contribution divergence,'' but I do not agree. `Contribution divergence' is not an explanation, but just another way of describing the observation.

Second, the authors have identified a player's utility with the earnings (line 463). However, when describing the experimental results, they mention risk-willingness (line 232), which implies that a player's utility could depend on his or her risk attitude. I guess that even their main observation that people become more favorable to cooperation in an uncertain environment could be described better by incorporating the risk attitude in utility properly.

6. PLOS authors have the option to publish the peer review history of their article (what does this mean?). If published, this will include your full peer review and any attached files.

Reviewer #1: **Yes: **Isamu Okada

Reviewer #2: No

---

## [Author Response · Author response to Decision Letter 1]

28 Dec 2024

Response to Editor comments:

We are very pleased to know that the two reviewers agree with you that the paper has good merit and addresses a highly relevant issue in behavioural evolution. Below, we provide a detailed point-by-point response to the reviewers' comments. In addition to addressing the reviewers’ comments, the following measures were also undertaken to comply with the PLOS ONE submission guidelines:

- We added an acknowledgement section to include the assistance of the local collaborators in line 641.

- We reviewed the reference list to ensure that it is complete and correct. According to scite.ai, there are no retracted articles in the reference list.

- The manuscript has been checked to meet PLOS One’s formatting requirements, including file naming and style, using the PLOS One’s LaTeX template.

- We added the experiment data together with the simulation source code in the publicly available repository, as well as described our data sharing plan in the additional information of the PLOS One’s submission form.

- All figure files have been checked and they comply with PLOS requirements through the Preflight Analysis and Conversion Engine (PACE) digital diagnostic tool, The png files were converted to tif and we addressed the issues reported by PACE.

Furthermore, we fixed the following typographical errors:

- Line 43: “cooperatove” to “cooperative”

- Line 57: “adressing” to “addressing”

- Line 102: “homogenous” to “homogeneous”

- Line 143: “experiement” to “experiment”

- Line 270: “perticipant’s” to “participant’s”

- Line 430: “commitee” to “committee”

We hope that the revisions have improved the manuscript to fully meet PLOS ONE’s publication criteria. Thank you again for this opportunity to revise our work.

Responses to Reviewer #1

Comment 1: "I think one of the key elements of this study is to test the three different types of uncertainty about the disaster (the probability of a disaster, threshold level, and impact). Therefore, I believe it is important to show the influences of them on the experimental performances. Although in Results, the authors mentioned contributions, contextual differences, responses to checks, and conditionality, the summary (figures) of results, especially the impact on the three different types of uncertainty, may be helpful for readers to understand."

Response 1: The main result about the impact on cooperation of the different uncertainty types is captured by comparing contribution levels, which are clearly presented in Figure 2A and 2B. Thus, we are not sure that we correctly interpret this comment. But there are two results: (1) about the general effect on uncertainty compared to a certain environment, and (2) individual differences in contributions between the uncertainty types. To clarify this, we divided the Contributions section (under Experiment Results) into "Effect of uncertainty" (line 197) and "Effect of different uncertainty types" (line 212). Grouping the findings in these subsections can make it more helpful for readers to understand the results and figures.

Comment 2: "The experiments were conducted during various periods in 2012–2013. I wonder why it took more than ten years to submit this manuscript. Would you explain the situation briefly?"

Response 2: We appreciate this query regarding the timeline. The long gap between conducting the experiments and journal submission was due to a combination of circumstances. These are: change of research focus and job priorities, as well as incorporation of additional multimodal analyses, particularly the simulation models. This is to ensure robustness of the results and incorporate new perspectives to enhance the overall quality of the paper. We provide transparency about this timeline in the Methods, and we can include this explanation if you and the editor think it is necessary.

Comment 3: "Their simulation results support their experiments. Although the source code is available, I don't understand the detailed setting of the simulation. Please add the explanation."

Response 3: We are not sure what is meant by "detailed setting", but we have added a description to the README-file in the GitHub repository where the simulation details are explained, as well as added code documentation to the source code.

Responses to Reviewer #2

Comment 1: "In Abstract, the authors say that This is explained by a contribution divergence,' but I do not agree. Contribution divergence' is not an explanation, but just another way of describing the observation."

Response 1: Thank you for this observation and we agree that "contribution divergence" is not an explanation but another way of describing the observation. We revised the Abstract, changing “This is explained by” to “In other words, we observe” a contribution divergence (line 14).

Comment 2: "The authors have identified a player's utility with the earnings (line 463). However, when describing the experimental results, they mention risk-willingness (line 232), which implies that a player's utility could depend on his or her risk attitude. I guess that even their main observation that people become more favorable to cooperation in an uncertain environment could be described better by incorporating the risk attitude in utility properly."

Response 2: We appreciate this comment. Game-theoretically, the utility consists of the monetary earnings alone. However, the risk-willingness naturally affects cooperation in uncertain/risky environments. To incorporate and clarify this, we added the following to the Contributions section under Experimental results (lines 206-211):"That contributions are higher when there is a risk of losing earnings (unless a contribution threshold is met) is both intuitive and consistent with the Nash equilibrium analysis in the Materials and methods section. However, the extent to which contributions increase in the presence of a disaster risk naturally depends on the participant's risk-willingness. We see evidence of this in the Contextual differences section below."

This is made clear in the Philippines-Sweden comparison where Filipinos had a higher level of self-reported risk willingness, and thus also contributed less than Swedish participants. Making the player’s utility depend on individual risk-willingness will make game-theoretical predictions complicated and will also require arbitrary assumptions on how risk-willingness affects utility.

---

## [Decision Letter · Decision Letter 1]

24 Jan 2025

Cooperation in the face of disaster

PONE-D-24-39170R1

Dear Dr. Luistro Jonsson,

We’re pleased to inform you that your manuscript has been judged scientifically suitable for publication and will be formally accepted for publication once it meets all outstanding technical requirements.

Kind regards,

The Anh Han, Ph.D.

Academic Editor

PLOS ONE

Additional Editor Comments (optional):

Reviewers' comments:

Reviewer's Responses to Questions

**Comments to the Author**

1. If the authors have adequately addressed your comments raised in a previous round of review and you feel that this manuscript is now acceptable for publication, you may indicate that here to bypass the “Comments to the Author” section, enter your conflict of interest statement in the “Confidential to Editor” section, and submit your "Accept" recommendation.

Reviewer #1: All comments have been addressed

Reviewer #2: All comments have been addressed

2. Is the manuscript technically sound, and do the data support the conclusions?

Reviewer #1: Yes

Reviewer #2: Yes

3. Has the statistical analysis been performed appropriately and rigorously? 

Reviewer #1: Yes

Reviewer #2: Yes

4. Have the authors made all data underlying the findings in their manuscript fully available?

Reviewer #1: Yes

Reviewer #2: Yes

5. Is the manuscript presented in an intelligible fashion and written in standard English?

Reviewer #1: Yes

Reviewer #2: Yes

6. Review Comments to the Author

Reviewer #1: I have confirmed all of the author's responses and agree to publish the article as is. I think this paper is of high quality.

Reviewer #2: (No Response)

7. PLOS authors have the option to publish the peer review history of their article (what does this mean?). If published, this will include your full peer review and any attached files.

Reviewer #1: **Yes: **Isamu Okada

Reviewer #2: No

---

## [Editor Report · Acceptance letter]

PONE-D-24-39170R1

PLOS ONE

Dear Dr. Luistro Jonsson,

I'm pleased to inform you that your manuscript has been deemed suitable for publication in PLOS ONE. Congratulations! Your manuscript is now being handed over to our production team.

Kind regards,

on behalf of

Dr. The Anh Han

Academic Editor

PLOS ONE